# Staff perspectives on fall prevention activities in long-term care facilities for older residents: "Brief but often" staff education is key

Neah Albasha[1,2]*, Catriona Curtin[1], Ruth McCullagh[3], Nicola Cornally[4], Suzanne Timmons[1¤]

1 Centre for Gerontology and Rehabilitation, School of Medicine, University College Cork, Cork City, Ireland, 2 Department of Rehabilitation Sciences, College of Health and Rehabilitation Sciences, Princess Nourah bint Abdulrahman University, Riyadh, Saudi Arabia, 3 Discipline of Physiotherapy, School of Clinical Therapies, University College Cork, Cork City, Ireland, 4 School of Nursing and Midwifery, University College Cork, Cork City, Ireland

¤ Current address: Centre for Gerontology and Rehabilitation, St Finbarr's Hospital, Cork City, Ireland
* neahalbasha@gmail.com

## Abstract

### Introduction

Falls are a serious health problem in long-term care facilities (LTCFs), affecting more than 50% of residents. A key role of LTCF staff is to assess fall risks and implement fall prevention activities. Understanding the barriers and facilitators is key to successful implementation.

### Methods

This descriptive qualitative study involving four LTCF facilities (varied provider types and sizes) in southwest Ireland. We recruited a convenience sample of 17 LTCF staff, who participated in semi-structured online 1:1 interviews (n = 7) or small group interviews (n = 10). The data were analysed using Braun and Clarke's reflective thematic analysis.

### Results

The participants included two directors of nursing, three therapists, one ward manager, one general practitioner, five nurses and five healthcare assistants. Six main themes were identified, reflecting factors that influenced fall prevention: a need for sufficient staff and appropriate skill mix; fall policy, documentation and leadership; equipment and safe environments; person-centred care; staff knowledge, skills and awareness in falls prevention; and staff communication and collaborative working. A wide range of approaches that supported LTCF staff to overcome barriers were identified, including audits and feedback, falls champions, fall prevention leaders, daily communication (e.g., safety pauses) and staff collaboration. Formal multidisciplinary meetings and identification systems to highlight residents at high risk of falling were not considered helpful. Staff suggested that education should be briefer, ongoing and practice-based ("brief but often") to promote ownership and responsibility.

**Data Availability Statement:** Data cannot be shared publicly because participants were interviewed, with interviews transcribed verbatim. Our data are qualitative and involve sensitive

information, and we must follow the policy of our research ethics board in UCC. It would therefore be inappropriate for this to be made public,as participants could be identified by the context of their interview responses (even though data is anonymised), which may include some overt criticism of the health service or local management structures. For requests or questions for researchers who meet the criteria for access to confidential data, contact Christopher Walsh, administrator and manager at the Centre for Gerontology and Rehabilitation, School of Medicine, University College Cork, via email: ChristopherWalsh@ucc.ie, or phone: 021-462-7347.

**Funding:** This current study was conducted as a part of a PhD project for the first author, funded by Princess Nourah bint Abdulrahman University, Riyadh, Saudi Arabia (Grant Number: Not applicable).

**Competing interests:** The authors have declared that no competing interests exist.

## Conclusion

LTCF staff identified several approaches to prevent falls in LTCFs as part of usual care, rather than lengthy, formal meetings and training. The potential role of families in fall prevention was under-appreciated and should be investigated further.

## Introduction

Falls are a major health problem in older persons. Globally, falls are the second leading cause of death, resulting in approximately 684,000 deaths yearly [1]. Falls can have physical and psychological consequences, such as fractures and fear of falling [2, 3]. According to estimates, the annual cost to the Irish economy of hospitalising older people with fall-related injuries is €59 million [4]. Fall rates in long-term care facilities (LTCFs) are approximately three times higher than that in community-based settings, with an estimated 1.7 falls per resident-year [5, 6]. Most LTCF residents are frail and vulnerable, with co-morbidities and disabilities [7]; thus, up to half experience falls annually, and 40% fall recurrently [8, 9]. Additionally, 10 to 25% of falls cause serious injuries, compared to 5% in the community [5, 6], with fractures occurring in 3–5% of residents yearly [2].

Many factors cause falls among the elderly. These can be classified into intrinsic factors (such as ageing, medical conditions, cognitive impairment, etc.), extrinsic factors (such as environmental hazards and medications) and psychological factors (such as fear of falling) [5, 10, 11]. These factors can be readily identified, and some are modifiable [5, 11, 12]. Many fall prevention interventions exist [6, 8, 13, 14]. Multifactorial interventions have been proven to reduce falls in LTCFs, whereas single-intervention effects are inconsistent [6, 8, 13].

Fall prevention is an important clinical quality indicator [15]. LTCF staff play a significant role in identifying residents at increased risk of falling, assessing specific falls risks and implementing fall prevention measures [4, 16]. A multidisciplinary team (MDT) effort is needed to prevent falls [17]. Reducing the burden of falls, improving residents' outcomes and facilitating changes in LTCFs require staff to become proactive rather than reactive [16–18]. Continuous professional development is vital for staff to change their behaviour, linked to enhanced knowledge and skills, while meeting residents' and providers' needs [19, 20].

However, given the complexity of falls, frailty, morbidity and the functional capacity of residents, staff encounter several barriers to fall prevention interventions [17]. In 2017, a systematic review (SR) of eight mixed-methods studies found 17 facilitators and 27 barriers in daily practice [21]. Most occur at social and organisational levels, while resident and economic/political levels were less influential. The most cited facilitators were communication and equipment availability. Barriers included staff feeling overwhelmed, helpless, frustrated and under-confident in their ability to manage falls, along with staffing issues, limited knowledge and limited skills (e.g., general clinical skill deficiencies), and poor communication [21]. It is necessary to identify techniques to overcome these barriers [21, 22].

This study was part of a wider mixed-methods exploration of falls prevention in LTCFs [23, 24], with the following key objectives:

- to explore the general perceptions of LTCF staff regarding fall prevention

- to explore perceived barriers in preventing falls, and suggested solutions

- to explore the facilitators that may support staff in fall prevention

## Materials and methods

### Study design

This research involved a descriptive qualitative study design. Reporting is based on the Consolidated Criteria for Reporting Qualitative Research (COREQ) [25].

### Setting

In the southwest Irish counties of Cork and Kerry, which share residential care governance and funding, 13 LTCFs for older people (from a total of 71) had participated in our previous survey and were eligible for inclusion [23].

**Sampling of sites.** A sampling framework was used, based on provider type, facility size (under/over 50 beds) and location (urban/rural). Five sites were selected; two private, two public and one voluntary. Four were in Cork (geographically larger and more densely populated than Kerry) and one in Kerry. Three were in urban locations; three were large.

### Participation selection

**Sampling of participants.** The primary researcher (NA) contacted the LTCF directors of nursing (DONs) by email and shared study information, asking for site participation in interviews, and a site champion to help co-ordinate the planned interviews: up to two 1:1 interviews, and one group interview (GI) per site.

Participants were employed full-time or part-time for at least three months at the site to be eligible. A purposive sample of LTCF staff roles was sought:

- A 1:1 interview was offered to DONs, ward managers, physiotherapists (PT), occupational therapists (OT) and visiting general practitioners (GP), etc., where participants occupied a senior or unique role.

- GIs were offered to staff nurses and healthcare assistants (HCAs), involving 4–6 participants, to reduce the influence of persons in authority, such as DONs and ward managers, and to increase their confidence in discussing barriers.

All targeted participants received invitation emails with study details, along with a consent form. All interviews were conducted online using Microsoft Teams. Participants received two email reminders to remind them prior to the scheduled interview, and to obtain their signed informed consent if not already supplied.

### Data collection

A semi-structured interview guide was developed (see S1 File). Several probing questions were asked to stimulate broader discussion. NA moderated all interviews. No prior relationship existed between participants or sites and the researcher. Digital recordings were made using Microsoft Teams. The 1:1 interviews lasted 17–40 minutes, while GIs lasted 30–60 minutes. Recruitment took place from June 2022 to December 2022.

Interview recordings were transcribed verbatim by NA and checked for accuracy. Data saturation was achieved after four 1:1 interviews and two GIs; however, additional interviews/focus groups were conducted to confirm emerging findings and achieve LTCF staff role diversity in order to enhance transferability.

## Analysis

We used NVivo software (QSR International) Version 2021 to organise the qualitative data and applied reflexive thematic analysis (TA) based on Braun and Clarke's six-stage analytical process [26–28]:

- *Data familiarisation*: Listening to recordings, reading/re-reading transcripts, taking notes on initial insights.

- *Coding the data*: The transcripts were divided into meaningful units and labelled with open codes, giving equal weight to all information. As transcripts were word-rich, the codes tended to be phrases, to maintain contextual information and capture meaning. As coding rounds progressed (see below), highly similar codes were combined.

- *Generating initial themes*: The data were categorised according to meaning patterns derived from the text's content. Considering the codes, reviewing the original data, and using notes, NA produced a set of themes and subthemes.

- *Reviewing and developing themes*: This involved iterations between codes and themes/sub-themes, choosing quotations to illustrate concepts, and assessing the relationship between ideas and patterns.

- *Refining, defining and naming themes*: This was performed using a Microsoft Excel spreadsheet. We extracted codes, subthemes and themes to a spreadsheet, to identify overlaps, thus clarifying themes, subthemes and standalone ideas.

- *Producing the report*: After reflection and feedback from all research team members, engagement with the literature and revision, a final thematic structure was developed.

## Trustworthiness

Four criteria were used to establish trustworthiness: dependability, confirmability, credibility and transferability [29].

- *Dependability*: The research team developed the methodology, interview guides and research questions. NA conducted a mock GI with three experts in qualitative research to obtain input on the clarity of questions, feasibility and factors influencing participants' views. Memos and annotations were used to ensure audibility.

- *Credibility*: Notes were recorded by the researcher following each interview, detailing participant interaction, body language and facial expressions. During GIs, we used member checking, seeking participants' opinions regarding other participants' perspectives. NA coded all transcripts; a second researcher (CC) independently coded two manuscripts. These independent codes were compared, using reflective notes where available, to ensure reliability and validity, and to cultivate reflexivity (two coding rounds). The refinement of themes through reflection was supported by a senior researcher (ST; 3 rounds) for further validation.

- *Confirmability*: During interpretation, supporting verbatim quotes from participants were gathered. As part of the interpretation process, all researchers gave input.

- *Transferability*: This was achieved by describing the data collection process and reporting on the range of staff roles involved, supported by data saturation and providing rich description of the findings [30].

### Ethics

The study was approved by the Social Research Ethics Committee (SREC) at University College Cork (UCC) (approval number: Log 2022–023). Written, informed consent was obtained from all participants. The study's goals were explained to all participants, who understood that participation was voluntary, that they could review the transcript and amend or delete sections and that they could withdraw at any time up to two weeks post-interview, without explanation. Participants were informed of the means of ensuring confidentiality, such as the coding of quotes (e.g., GI 1, PT1, etc.). Anonymised transcripts were stored on university hard drives which were password-protected.

## Results

### Participant characteristics and general views on falls in LTCFs

Two public sites, one private site and one voluntary site participated. Three were large, and one was small, with three located in Cork, one in Kerry. Seventeen LTCF staff (15 female, two male) participated in total, across three GIs (n = 10; five nurses and five HCAs) and seven 1:1 interviews. Almost all had experience of working with older people in LCTFs for 11+ years. The seven 1:1 interviewees involved two DONs, two PTs, one ward manager (Ward M), a therapy manager (Therapy M) and a GP.

Generally, opinions varied on whether falls could be prevented. Some believed falls were inevitable: *"Even [. . .] all the things we put in to prevent the falling, they still fall by the time we get to them; they're on the floor whatever we do"* (**HCA, GI 3**). However, most thought falls could be reduced: *"I suppose there are some falls that can be prevented, others not"* (**DON1**); *"the falls risk can be reduced but cannot be fully prevented"* (**Ward M**). Participants cited several reasons why falling cannot be completely prevented: residents' ages, frailty, cognitive ability, level of dependency and belief in their abilities. Some stated falls are complex, caused by multi-faceted factors: *"We know falls are multi-factorial. You can have a variety of factors, both physical, intrinsic, and extrinsic factors, that produce falls"* (**DON2**). Fluctuation in patient status was also problematic: *"Sometimes, the nature of patients can change over the course of the day if they become unwell, or if you have somebody with dementia [. . .] maybe something happens over night or during the day, and their function changes"* (**PT1**).

### Factors that can reduce falls

These are organised under **six key themes (Table 1)**.

**Theme 1: The need for sufficient staff and appropriate skill mix.** This was the most common theme discussed in all interviews. It involved two subthemes: staffing levels and appropriate skill mix, and access to therapists.

*Staffing levels and appropriate skill mix.* According to participants, increasing staffing levels and maintaining an appropriate staff-to-resident ratio are key to preventing falls in LTCFs. Staffing shortages and staff-to-resident ratios were commonly discussed in relation to the care needs of highly dependent residents, along with the provision of supervision. Falls are most common during night shifts and handover, due to low staffing levels.

*"You have only two nurses and one care assistant at night, so it is hard to be eyes everywhere and, in every room"* (**Nurse, GI 3**).

*"We have got a skeleton staff of maybe two nurses and two healthcare assistants per 20 or 30 patients, and in the night, we do not have enough staffing"* (**GP**).

**Table 1. Themes/subthemes within staff perspectives on factors that can reduce falls among older residents.**

| Themes | Subthemes |
|---|---|
| The need for sufficient staff and appropriate skill mix | • Staffing level and appropriate skill mix<br>• Access to therapists |
| Fall policy, documentation and leadership | • Audits and feedback<br>• Fall policy, risk registers, resources<br>• Supportive leadership and management |
| Equipment and safe environment | • A safe environment<br>• Assistive equipment and supplies<br>• Identification systems for residents at high risk of falling |
| Person-centred care | • Individualising care<br>• Assessing fall risk<br>• Knowing residents holistically<br>• Promoting residents' autonomy and independence<br>• Involving residents and families in fall prevention |
| Staff knowledge, skills and awareness in falls prevention | • Staff education and training<br>• Fall champions<br>• Staff motivation, self-efficacy and responsibility<br>• Practice-based learning |
| Staff communication and collaborative working. | • Staff communication and teamwork<br>• Multidisciplinary approaches |

Other staffing issues included staff movement within the LTCF, time constraints, staff workload, staff turnover due to the COVID-19 period and adverse effects from the work environment, such as back pain and lower interest in LTCF posts:

> *"I think there is a lot of [what] I would call COVID fatigue [...] the staff are tired, and so, for the first time here, we have had a lot of staff actually stopping work and giving up"* (**PT1**).

> *"I think people, especially after COVID, do not want to come into this job. We got the hard jobs [...] I think a lot of people have left this career"* (**HCA, GI 1**).

Some suggested agency staff were a barrier, due to unfamiliarity with residents and the environment: *"They give great care, but they just do not know the residents"* (**PT1**); *"Agency staff [...] do not know anything what is happening around the environment and around the residents"* (**HCA, GI 1**). Participants believed that more experienced staff at LTCFs better promoted fall prevention: *"They really need those staff who are more experienced there; they need a more long-term staff there who are aware of what is happening in the environment and the residents"* (**Nurse, GI 1**).

Two interviewees discussed inconsistent staffing levels and specialised care input between public and private facilities. The latter had low staffing levels and variable therapy input, with low resident-to-staff ratios during daytime/nighttime, leading to group supervision and the restriction of resident freedom:

> *"The [public] nursing homes have to have a certain number of carers per residents' numbers, and in private nursing homes, we do not have that. It is a huge problem"* (**PT1**).

> *"A lot of private nursing homes would have a tendency to [...] corral the residents grouped into one sitting room, which is awful [...] they [try to] increase supervision"* (**PT2**).

Staffing levels should be standardised and regulated in all LTCFs to ensure consistency:

"*[Public] nursing homes have that regulation in place, but not private nursing homes [. . .] to try and create an environment where we can have consistency across the board in all nursing homes, both public and private*" **(PT1)**.

The potential cost-balance in terms of the burden of fall-related consequences was discussed by one:

"*we need to resource [LTCFs] so that you have got adequate staffing levels to keep patients safe and eventually save money, because not having a patient in accident and emergency and going for surgery for a fractured femur will save an awful lot of money*" **(GP)**.

*Access to therapists*. Most participants cited insufficient therapy input in LTCFs as a barrier, particularly PTs and OTs, especially in private LTCFs:

"*In most nursing homes, we have a physio service for 2/3 hours, once a week at most, and it is just not sufficient. You cannot follow a falls programme or fall prevention system with just three hours of physio input*" **(PT1)**.

"*I think at least two physiotherapists to look after those residents, because they need an assessment [. . .] Out of 110 residents, we have only one physiotherapist, and sometimes, if she is on holiday, we will wait for two weeks or one week until she comes back*" **(HCA, GI 1)**.

Not all sites had this issue: "*We have a physiotherapist on site, and we have a lot of occupational therapists that link them with a physio, so that would not [be a problem in] this home, but I could understand why it is to others*" **(DON2)**.

Most participants recognised the value of PT/OT roles, discussing how they assist with walking aids, equipment and ergonomics during assessment and care planning. Therapists were seen to deal with the frailty (mobility) of LTCF residents, whereas nursing and support staff mainly focussed on daily nursing care: "*the OT is probably more valuable to us, because [our patients] are so physically frail*" **(Ward M)**.

The role of therapists as educators and trainers was stated by one:

"*I do think physiotherapists and occupational therapists have an important role [concerning] staff education and training. I do not think it's as much about hands-on physiotherapy as it is about education and training for staff*" **(GP)**.

Participants had some suggestions for augmenting PT/OT roles via assistants, whether "*a physiotherapy assistant*" **(GP)** or the participation of healthcare assistants in physio sessions: "*Demonstrating, like, to join the session with care staff, to show, actually, this can work, and it will have [longer-term] benefits*" **(PT2)**.

**Theme 2: Fall policy, documentation and leadership.** Most participants discussed these as factors to reduce falls, under three subthemes: audits and feedback; fall policy, risk registers, resources; and supportive leadership and management.

*Audit and feedback*. This topic was discussed extensively by all seven interviewees and one GI, and was considered a crucial strategy to facilitate fall prevention. Views included the role of auditing in providing a visual guide, as a quality care indicator, and in directing change via action plans. Audits were perceived to give insight into compliance with national and international guidelines for fall prevention and to raise staff awareness, empowering them to take responsibility. Additionally, audits were considered to play an important role in increasing family awareness, especially for residents in private rooms:

"*I think audits are important [and have] highlighted certain areas that we have been able to then alter*" **(PT1)**

"*I think clinical audit[ing] is always helpful, because that is a snapshot of the way we are in clinical practice [. . .] patients, staff, doctors and physios [should] reflect [on] what is happening on their own unit, how that reflects national and international guideline[s] [. . .] I think audit[ing] is a really valuable tool in order to improve the quality of care that we deliver*" **(GP)**.

A few discussed the importance of audit feedback:

"*The audits are only as good as the feedback [. . .] if we know there is something that we are falling down on, we can improve*" **(HCA, GI 2)**.

"*Learning from the recent audit of falls. You can see patterns [. . .] Is there a pattern of where these falls are happening? I think it's always more concrete*" **(DON 2)**.

*Fall policy, risk registers, resources*. The importance of having a **good fall policy**, especially on procedures post-fall, was mentioned by a few. One highlighted the value of "fall packs", which contain items for supporting staff who witness falls (e.g., "two-minute rule" assessments and flowcharts for actions post-assessment and items to support staff in safely transferring fallers), serving as reminders and offering practice-based learning approaches that enhance their ownership and responsibility.

"*I think the system we have here is working [; it] is called a fall pack, so when a resident falls, there should be a bit of a fall pack near a nurse's station [. . .] which contains a sling for someone to be hoisted. It contains a two-minute rule and a flowchart that comes with it, and other information [. . .] this pack just allows little quick reminders [. . .] telling you which direction you can take your path and getting assistance with it*" **(PT1)**.

A **GP** discussed the importance of **fall risk registers** to support staff being proactive in identifying residents at high risk of falling and to demonstrate commitment to fall prevention: "*I think this is important medically and legally, because, I suppose, number one, we have to do the right thing, and, number two, we have to be seen to do the right thing, because it will happen; some patients inevitably fall. We have to show that we have tried our best*".

Another described using a national awareness-raising tool with colour-coded days each month to indicate falls among residents. **DON1** stated, "*so, we have a red [a fall occurred] and green [no falls occurred] day, so, if we have a fall, then we initiate, then, a risk response to that [. . .] How can we put in additional features to improve or to try and decrease the risk of further falls?*"

*Supportive leadership and management*. Some participants highlighted the importance of leadership and supportive management, which impact staff performance in preventing falls. Leaders can embrace and facilitate creativity, implementing quality care metrics and sharing research projects. They can enhance LTCF staff responsibility and empowerment, particularly for non-management staff, by facilitating staff peer learning and encouraging staff to obtain continuous training to improve their skills.

"*I think leadership is important. From the nursing staff to CNM to the director of nursing, [we] really have to drive change [. . .] If you do not have leaders looking at pushing new ideas and pushing forward, like taking part in a research study, being involved in quality care*

*metrics and going to study days, and then coming back and feeding that back to the staff, encouraging staff to go to training"* **(Ward M)**.

However, another participant felt that management had a negative mindset, leading to resident immobility and dependency, due to excessive focus on safety and bed capacity: *"I received a very negative comment [. . .] The problem is most of the [managers] are not interested in it. They are only concerned that the rooms are full. They do not think about pro-mobility or the quality of the care"* **(Therapy M)**.

One perceived a barrier in management hierarchies, speaking of a management to staff-on-the-ground gap and bureaucracy from middle layers. *"I think, sometimes, there might be a barrier between, maybe, management and what the staff are witnessing on the floor [. . .] but then, I think, sometimes, with the middle management [. . .] there might be a delay in that, so sometimes, we were not sorted straightaway"* **(Nurse, GI 2)**.

**Theme 3: Equipment and a safe environment.** This theme included three subthemes: a safe environment, assistive equipment and supplies, and identification systems for residents at high risk of falling.

*A safe environment.* Several participants discussed barriers such as the building's age and geographical location. Small rooms and non-ensuite rooms were felt to increase fall risks. Private rooms are more difficult to monitor than multi-occupancy rooms; the latter enabled supervision, and other residents could provide support, enhancing safety. Other environmental hazards included flooring, clutter, room layouts, etc. Suggested solutions included de-cluttering, lowering bed levels, providing good lighting, having appropriate furniture, etc.

*Assistive equipment and supplies.* Some participants mentioned appropriate equipment such as walking aids, call bells, crash mats and sensor mats. One highlighted the cost of these: *"funding in relation to equipment [. . .] I definitely think that I go back to HSE and finance"* **(DON1)**. Participants discussed sensor mats increasing staff alertness as a facilitator. However, barriers included the overuse of sensor mats; supporting residents with toileting needs were complicated; and alarms sometimes did not receive a quick response due to low staffing ratios and time constraints. The use of sensor mats was also perceived to restrict residents' movement and make them agitated, resulting in falls. Participants suggested that sensor mats should be distributed only to those in need, not everyone, with five or fewer sensor mats in the entire unit, with staff educated on their use.

*Identification systems for residents at high risk of falling.* All participants agreed that colour-coding, bracelets, etc., for identifying residents at high risk of falling was not the best approach to helping staff prevent falls. It was claimed that the Health Information and Quality Authority (HIQA; an independent, state-appointed regulatory body) did not support these: *"We do not have a display for [fall leaves [high-risk logo]] here with the residents. I think that's good in a hospital setting [. . .] but I do not think HIQA really embraced it"* **(DON2)**.

It was also noted that identification systems lead to staff complacency, and room-based systems can quickly get out-of-date:

*"I find that a lot of the poster system and the coloured leaves system do not get changed between residents or residents moving rooms [. . .] I think it potentially makes a nursing home look good, but the practicality of implementing it fails very quickly through complacency"* **(PT1)**.

Small sites said that all staff were already familiar with residents, so fall-risk identification systems only compromised the dignity (privacy) of residents.

**Theme 4: Person-centred care.** This theme focussed on providing individualised assessment and care for fall prevention, as well as relationships between staff and residents/their families. Four subthemes related to care provision: individualising care; assessing fall risks; knowing residents holistically; and promoting residents' autonomy and independence.

*Individualising care*. A perceived barrier was that care was overly task-oriented. Participants considered person-centred, individualised care to be the best approach to preventing falls: *"we're really trying to push back to becoming more person-centred"* (**PT1**).

A key activity was **appropriate supervision and monitoring** of residents at high risk, whether by placing them near a nurse station or visible area, providing hourly check-ups or establishing regular toileting routines: *"we do move the patient or resident at high risk for it near to the nurse's station, keep eyes on in the resident–just, like, more observation, monitoring the resident"* (**Nurse, GI 3**). Providing supervision during movement and mobility was also indicated for residents newly admitted to LTCFs to better understand their risk factors. Supervision should be increased once they identify a unit at higher risk of falls or a resident with the potential for adverse medication effects, especially at night, within the first few hours of taking the medication. The role of 1:1 care was discussed: "in certain cases in the site, they might need to be specialised, as in a one-to-one, if they were very acutely confused and very distressed" (**PT2**). Another stated:

> "W*e noticed that in our dementia unit that was at high risk of falls, and we looked at them increasing the supervision in this area, and that has less amounts of falls in that area; there was a highly active unit"* (**DON2**).

*Assessing fall risks*. Using a proactive approach to identify potential risk factors for each resident was considered key to preventing falls and providing appropriate care. LTCF staff should perform ongoing/daily risk assessments:

> *"the important thing is identifying what is causing the falls for each individual resident; it could be different with each resident, not the same for everyone"* (**Nurse, GI 3**).

> *"It is a daily assessment, ongoing assessments, actually; we are constantly assessing them"* (**HCA, GI 2**).

They should conduct a post-fall re-assessment to identify any additional risk factors and update their care plans. Participants also discussed the importance of conducting medication reviews, including "*the harmful effects of sedating medications*" (**GP**). They noted the importance of having appropriate footwear and clothing.

*Knowing residents holistically*. LTCF staff believed that knowing their residents extensively can prevent falls. This includes understanding their habits, activities and usual tasks, knowing their care needs, such as their medication and side effects, as well as their dependency level and personal fall risk factors:

> *"knowing your resident is really key and knowing their story and knowing their likes and dislikes"* (**DON2**).

> *"Knowing your residents; I mean, I have [XX] residents here with me in this particular environment, and I know them all individually. I know all their manual handling, and I know their daily care needs"* (**PT1**).

In addition, staff must know the whole medical history of new residents in order to become familiar with them. A few participants claimed that not knowing residents hindered staff

efforts to prevent falls in LTCFs, while one said that all staff were familiar with residents in their site, so it was not a barrier **(Ward M)**.

*Promoting residents' independence and autonomy.* A lack of resident activities and mobility in LTCFs was considered to increase the fall risk. There was much discussion on how improving residents' mobility, maintaining a healthy physical condition and using "positive risk-taking" might increase falls, but restricting movement was against their human rights:

> *"We advocate for independence as much as possible [. . .] our fall risk would be higher because of that"* **(PT1)**

> *"We have a huge philosophy of independence and encouraging mobility, so [. . .] we do take a lot of risks"* **(Therapy M)**

> *"We are balancing risk and autonomy [. . .] we promote patient mobility, but we have to be cognisant of the fact that by promoting mobility and independence and autonomy, they carry the risk of falls"* **(GP)**

*Involving residents and families in fall prevention.* A few participants indicated a lack of safety awareness among residents and poor communication between staff and residents as barriers to preventing falls. The best way to overcome these is by educating residents and their families.

Good communication with residents was also highlighted by another participant:

> *"People here in nursing homes do not communicate with residents. Communication is important. You have to get consent; you have to inform them"* **(Therapy M)**.

A few participants felt that involving families in care planning may prevent falls, since they can assist in providing care, doing exercise and improving activity levels:

> *"Family involvement is really important, because they can help them to do the exercise when we do not have time, and they can bring them, like, proper shoes"* **(HCA, GI 3)**.

Personal resident preferences were important, as *"some residents [. . .] would not want their families involved"* **(Ward M)**.

Conversely, family involvement in care plans was viewed by some as a barrier for LTCF staff, because their expectations may be unrealistic, placing negative pressure on staff. For example, sometimes, families requested bed rails or chair rails, which can cause residents to fall or be injured. Families often hindered residents' mobility due to fear of falling. It was also felt that it is more beneficial to involve families in fall prevention in the community than in LTCFs:

> *"A lot of the time, families are kind of removed from the reality of what is happening [. . .] they just have unrealistic expectations and [tell] me what to do"* **(HCA, GI 2)**.

Furthermore, there can be conflict within families (especially in large families) due to differing preferences.

**Theme 5: Staff knowledge, skills and awareness.** All participants discussed this theme, across four subthemes: staff education and training; fall champions; staff motivation, self-efficacy and responsibility; and practice-based learning.

*Staff education and training.* A lack of knowledge about fall risk factors, especially medication knowledge, was a barrier to preventing falls. Staff were perceived to sometimes lack the skills to manage residents with dementia or cognitive impairment with unsafe behaviours.

Furthermore, new staff may lack the knowledge and skills to perform their roles, which could lead to falls:

> "A *lot of new staff we have gotten in recent years don't have a proper training experience, because they were trained online*" **(HCA, GI 1)**.

> "N*ew staff members are not trained, or it takes time to learn about the resident*" **(Therapy M)**.

This lack of experience in some staff increases the workload and responsibility on experienced staff, leading to double-tasking and rushed work.

Training/education for staff was key to preventing falls, particularly through increased staff awareness of risk factors: "H*aving education and training of frontline staff, but as part of that education and training, incorporating practical strategies*" **(GP)**. This needed to be ongoing (yearly) education, *"it's ongoing education, really important"* **(PT1)**.

Participants noted the importance of education on specific topics such as fall assessment, transfer, manual handling and dementia training. Induction training for new staff should include manual handling techniques. Some suggested it should be mandatory:

> "*They [should] have a week or even three days of basic training on all different types of manual handling techniques, with different examples of different residents before they even get put onto the floor*" **(HCA, GI 1)**.

One participant **(PT2)** noted the value of educational posters reminding staff of factors leading to falls and how to control them.

*Fall champions*. Fall champions were strongly supported, working as trainers for all staff, creating a fall risk register, encouraging staff to implement training while monitoring and facilitating changes in fall-prone areas:

> "W*e have falls champions within the service; it does help to have people specifically interested and to drive change*" **(Ward M)**.

> "W*e have the senior care assistant as a fall champion; they know, like, how to react in case of emergency [. . .] that is really helpful*" **(Therapy M)**.

One participant claimed that fall champions are more effective than audits for staff learning through practice, as they offer real-time, continuous observation and reinforcement, while auditing is a time-based phenomenon:

> "*A fall champion is in real time, and a constant thing, whereas an audit is a position in time [. . .] once a month or every three months [. . .] it is falling flat in the time that you are not auditing. Fall champion–there's a lot more observation around falls, and it's more of a heightened topic when it's constantly going on*" **(DON1)**.

*Motivation, self-efficacy and responsibility*. According to some, a lack of LTCF staff ownership and perceived responsibility for fall preventions are barriers. Respondents felt that it is vital to promote staff ownership and responsibility for fall prevention training, since everyone is responsible for preventing falls:

> "*Our training here is encouraging staff to take ownership and responsibility. We are just trying to get people to be more responsive owning it and taking responsibility*" **(PT1)**.

*Practice-based learning approach.* Participants discussed this educational approach as one of the most effective: learning via practice, either informally or through collaboration. Moreover, staff were felt to learn continuously, particularly after each post-fall incident. Collaborative learning was emphasised via strategies implemented daily, such as safety pauses/huddles (discussed later) and fall champions. Participants indicated that they learned from real-life stories of residents. Through these approaches, staff were made aware of residents at high risk of falling, and their knowledge and practice were continually refreshed:

"*I think that is why I have the safety pauses and champions going here, because it [. . .] keeps things to the top of the minds*" **(DON1)**.

**Theme 6: Staff communication and collaborative working.** This theme was discussed by all participants, across two subthemes: staff communication and teamwork; multidisciplinary approaches.

*Staff communication and teamwork.* Few participants reported poor communication between staff. Communication was an important facilitator in preventing falls ("*communication is very important to handover*" **(PT1)**), especially communication of changes in a resident: "*daily communication on any kind of upgrade or downgrade of a patient's, kind of, functional status*" **(PT2)**.

A **safety pause** is an effective strategy, as indicated by many participants, also known as a "huddle" or mini-meeting. These typically occurred after handovers, over 5–10 minutes, and were felt to enhance staff empowerment and responsibility, support peer learning, ensure patient safety and promote proactive approaches:

"*We would do safety pauses here, which have been quite successful*" **(DON 1)**.

"*Huddle meetings, which are just a quick five minutes, [. . .] work quite well [. . .] I think that the minute you empower somebody who is not management, they take it on*" **(PT1)**.

Apart from these group events, several participants noted how important it was to work as a team and to support each other by providing advice or guidance:

"*We are here to just support each other in that role, and I would expect someone to tell me 'You forgot this bit'. Then, I will acknowledge it and do it*" **(PT1)**.

*Multidisciplinary approach.* Collaboration between the wider MDT was considered important for all participants. They acknowledged that fall prevention needs to be multidisciplinary. However, participants considered a formal MDT meeting to be ineffective, as, typically, MDT meetings took place every 3–6 months and were deemed irrelevant unless resident assessments and conditions changed within that time frame. Furthermore, it was considered time-consuming to discuss a limited number of residents in depth, with some suggesting a safety pause could cover everything in MDT meetings more efficiently:

"*A lot of time [is] wasted in meetings [. . .] it's just ineffective use of time*" **(PT1)**.

"*We would have staff MDT meetings that we would raise certain things [. . .] But I suppose our daily safety pauses really take that*" **(DON1)**.

To be more effective, MDT meetings should be conducted daily or weekly, while MDT "on-call consultation" or informal meetings, such as walking meetings in corridors, were perceived as effective in LTCFs for interacting and sharing information:

"A *lot of our MDT conversations would happen in the corridors when we bump into people, or we would have walking meetings [. . .] that system seems to work in this particular LTCF*" **(PT1)**.

## Discussion

This study aimed to understand the barriers faced by LTCF staff and the perceived facilitators of fall prevention, which are essential to gaining insight into how fall prevention can be optimised, as staff are key to preventing falls. Although staff discussed their experience that falls were not completely preventable, and a few had pessimistic attitudes, most were motivated and positive.

Overall, LTCF staff provided a wide range of approaches (such as champions, audit feedback, practice-based learning, safety pauses, etc.). These are together described as "brief but often", as they are generally brief, targeted and used on an ongoing basis as needed, integrated into daily work. This contrasts with what staff considered onerous, i.e. fixed, sporadic "add-ons" to usual work, such as routine resident-by-resident MDT meetings; long, formal training events; or detailed but sporadic audits. As their preferred peer-learning and change-supporting activities are quick and can be easily integrated into daily routines, based on real-life scenarios encountered in units, they keep staff engaged and motivated.

Our findings indicated inadequate staffing levels, including access to therapists, poor staff-to-resident ratios, high workloads and little time. These, along with agency staff, were perceived as barriers to preventing falls. Inadequate staffing levels can increase organisational care burden [31] and impact resident safety and care [32–37]. Staffing levels inversely related to patient safety in some studies [17, 32]. According to an SR, inadequate staffing affected direct care, such as toileting assistance and monitoring [38]. The benefits of PT/OT input in preventing falls are well evidenced [39, 40]. According to the World Fall Guideline, physiotherapists are required to tailor intervention exercises and physical activity to individual residents [7]. Our results found that staffing levels are unequal across LTCF types, as consistent with an SR which found that physiotherapy staffing levels differ significantly across nursing homes [39]. The development of a quality standard for staffing levels should be examined, along with the relationships between staffing levels, care planning practices and fall rates. A framework to determine the number and skill mix of nurses and HCAs needed in each care area is being developed in Ireland, based on patient numbers and care needs. This is currently being piloted in non-acute care settings, including LTCFs [41].

Although insufficient staffing levels were perceived as a significant barrier to preventing falls in LTCFs, staff interviewed also believed that having experienced staff was key. Other studies have reported that although younger or less experienced staff members might have "fresh" knowledge and skills to prevent falls, they lack practical application and real-life experience [42], whereas fall-awareness, self-confidence and knowledge were significantly higher among those with extended experience [43]. Our linked knowledge survey of 155 respondents across 13 LTCFs showed that staff with the most clinical experience in working with older people (>11 years' experience) scored higher in fall prevention knowledge than those with two years' experience or less [23]. Experienced staff also have familiarity with residents and the clinical environment [44], allowing them to make appropriate clinical judgments [17]. For example, experienced staff may be more familiar with the preferences, habits and communication styles of residents, allowing them to identify potential fall risks in certain situations [32]. They may have a deeper understanding of cultural norms and the uniqueness of their facility's culture, supporting implementation of change [17].

Our findings showed that audits and feedback are perceived as helpful approaches for reducing falls. These assess the quality of current performance and guide change towards better care, where feedback for staff can influence their clinical practice [45]. A Cochrane review found that audits and feedback increased compliance with requested practices by over 4.3% in various clinical settings, most effective when (internal) supervisors or peers delivered them [46]. A descriptive clinical audit study involving 13 LTCFs in Australia similarly reported that LTCF staff involvement in auditing processes strengthens everyone's responsibility to translate evidence into practice [47]. Although there was no specific evidence to support audits/feedback in our SR of interventional studies [48], it is understood that audits and feedback, paired with supportive leadership, can reduce falls in a sustainable manner. Further research is needed on their particular impact within multifactorial interventions.

Our findings indicated that sensor mats are perceived as helpful for maintaining staff awareness of falls risk, but significant disadvantages include increasing resident agitation and interfering with other residents' care. Over-using sensors may lead to falls; staff education can support their proper use, and staff performance may be improved if unnecessary alarms are de-implemented [49]. Additionally, our findings revealed that using identification systems for residents "at high risk of falling" was unhelpful in LTCFs (where most are high-risk or are well-known to staff) and contrary to promoting residents' dignity (privacy). This contradicts a pilot study involving seven LTCFs, assessing acceptance of a fall risk management intervention involving 36 staff members [33], where a colour-coded system was reported as being easy to use and helpful, especially for rotational shifts.

Person-centred care provision may reduce falls, strengthen relationships between residents and staff and improve care outcomes [50]. It has been reported that staff are able to predict resident responses more effectively when they know them personally [51]. Communicating and educating residents, as well as respecting their independence and autonomy, and encouraging their participation in physical activity, were perceived to reduce fall incidents in the literature [52, 53]. In contrast, a small qualitative study conducted in Norway found that nursing staff in LTCFs focused more on fall prevention and protection than person-centred care approaches [42].

Involving residents and families in an effective and health-promoting manner was suggested in this study, such as families providing some personal care and improving resident activity levels, building on previous literature that families are vital to residents' overall wellbeing [54, 55] and that active involvement of residents and families as team members can improve communication around falls [35]. However, our participants did not want to involve families in healthcare planning, as their expectations might be unrealistic, placing negative pressure on staff, with examples given of families requesting bed rails or hindering residents' mobility due to fear of falling. A small qualitative study in LTCFs in Taiwan highlighted that fear of "blame" from families led to an over-focus on preventing falls (including the use of physical restraints) [53]. Further research with residents, their families and staff might lead to better understanding of how a balance can be struck that supports autonomy and person-centred care.

Staff education and training were perceived as vital, particularly for new staff, especially situationally-relevant and practice-based learning (PBL). Our findings reflected our previous survey results [23]. PBL is informal and reflective, contributing to professional understanding and responsibility, as decisions must be based on clinical judgment [44]. It provides opportunities to learn new skills, refine organisations and align public health systems with desired practices [56]. Within our study, fall champions and safety pauses/huddles were identified as key sources of PBL, although no evidence has indicated their effectiveness within a multifactorial approach [48].

In the workplace, fall prevention is a team effort; the main facilitating factor is staff communication [17]. This increases engagement and provides opportunities for continuous improvement of behaviour and professional development [51]. Our findings demonstrate the perceived importance of staff communication, reflecting a previous study with 41 nurses and HCAs in four LTCFs in Canada [35]. Professional collaboration is crucial for enhancing the quality of care and safety, empowering staff to take action and overcome implementation barriers, improving professional practice and health outcomes [57–59]. Our participants felt that staff teamwork and informal meetings were more effective than formal MDT meetings for reducing fall risks. Likewise, a previous study found that informal communication was perceived to play a significant role in highly collaborative, dynamic and information-rich clinical environments [51].

## Strengths and limitations

We gathered rich exploratory and explanatory information within a variety of LTCF providers, both urban and rural, across many different disciplines, gaining a comprehensive understanding from various perspectives. However, certain staff categories, such as GPs, ward managers and occupational therapists, were under-represented despite invitation. One site declined participation, and one interview group had only two participants, perhaps reflecting COVID-19 pandemic related increased workloads. Overall, data saturation was reached, and sufficient numbers of key staff (nurses and HCAs) were included, from four different sites.

## Conclusion

Six main themes were identified in this study as key factors in fall prevention in LTCFs: the need for sufficient staff and appropriate skill mix; fall policy, documentation and leadership; equipment and safe environments; person-centred care; staff knowledge, skills and awareness in fall prevention; and staff communication and collaborative working. A variety of approaches to integrating fall prevention into daily routines were suggested to make it more feasible and practical ("brief but often"), including rapid-cycle audits/feedback, fall champions and leaders, daily informal communication (e.g., safety pauses) and staff collaboration. Lengthy, formal MDT meetings and education sessions, and identification systems to identify residents at high risk of falling, were deemed unhelpful. All of these approaches need to be considered in fall prevention policy and practices. In addition, despite staff wariness of a potential over-focus on fall prevention versus resident autonomy, the involvement of families in fall prevention requires more exploration in terms of a potentially helpful role in supporting care and exercise.

## Supporting information

**S1 File. Interview semi-structured guide.**
(DOCX)

## Acknowledgments

The authors are grateful to the directors of nursing at all four participating sites in Cork and Kerry, who allowed us to recruit participants for this study and facilitated data collection.

## Author Contributions

**Conceptualization:** Neah Albasha, Ruth McCullagh, Nicola Cornally, Suzanne Timmons.

**Data curation:** Neah Albasha.

**Formal analysis:** Neah Albasha, Catriona Curtin, Suzanne Timmons.

**Methodology:** Neah Albasha, Ruth McCullagh, Nicola Cornally, Suzanne Timmons.

**Writing – original draft:** Neah Albasha.

**Writing – review & editing:** Neah Albasha, Ruth McCullagh, Nicola Cornally, Suzanne Timmons.

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
