## [Decision Letter · Decision Letter 0]

7 May 2024

PONE-D-23-40350"Brief but often": Staff perspectives on implementing falls prevention activities in long-term care facilities in older persons: a qualitative studyPLOS ONE

Dear Dr. ALBASHA,

Thank you for submitting your manuscript to PLOS ONE. After careful consideration, we feel that it has merit but does not fully meet PLOS ONE’s publication criteria as it currently stands. Therefore, we invite you to submit a revised version of the manuscript that addresses the points raised during the review process.

We look forward to receiving your revised manuscript.

Kind regards,

Nabeel Al-Yateem, PhD

Academic Editor

PLOS ONE

3. In the online submission form, you indicated that [Anonymous data used and analysed during this study are available from the corresponding author for non-commercial projects on reasonable request.]. 

Reviewers' comments:

Reviewer's Responses to Questions

**Comments to the Author**

1. Is the manuscript technically sound, and do the data support the conclusions?

Reviewer #1: Yes

2. Has the statistical analysis been performed appropriately and rigorously? 

Reviewer #1: N/A

3. Have the authors made all data underlying the findings in their manuscript fully available?

Reviewer #1: Yes

4. Is the manuscript presented in an intelligible fashion and written in standard English?

Reviewer #1: Yes

5. Review Comments to the Author

Reviewer #1: I think it is very well written. Please address the following minor points.

1. Lines 238-240 suggest that having experienced staff reduces the risk of falls. Specifically, what is the difference between younger and more experienced staff? I would like to see a few sentences in the discussion to include perspectives from previous studies.

2. line573

Topic 5 is probably a mistake for topic 6.

6. PLOS authors have the option to publish the peer review history of their article (what does this mean?). If published, this will include your full peer review and any attached files.

Reviewer #1: No

---

## [Author Response · Author response to Decision Letter 0]

20 May 2024

Response to Reviewers 

Response: Thank you for reviewing our manuscript. We have formatted all headings in the manuscript, from the abstract to the supporting information section after the reference list, according to the journal's guidelines. This includes using Heading 1, 2, 3, and so on, as specified by the formatting rules. In the results section, heading level 4 has also been included for all subthemes so as to reduce the use of italics. Additionally, we have adjusted the formatting of the manuscript’s title to comply with the journal's style requirements. Across the entire text, all changes are highlighted in red.

Response: Thank you for your note and for highlighting the potential citation advantage of depositing data in a repository. However, our data are qualitative and involve sensitive information and we must follow the policy of our research ethics board in UCC. 

3. In the online submission form, you indicated that [Anonymous data used and analysed during this study are available from the corresponding author for non-commercial projects on reasonable request.]. 

Response: As stated above, due to the nature of the data, we are happy to provide data to others on a case-by-case basis, where we can limit or redact data as needed, to maintain confidentiality. However, we cannot have the full data set publicly available, as participants could be identified by the context of their interview responses (even though data is anonymised), which may include some overt criticism of the health service or local management structures. Participants consented to participation and were very honest in their responses based on the data being shared as we described in the PIL, and our ethics committee approved this protocol, thus we cannot now change to deposition in a repository, or inclusion within the main body or appendices of the paper, as this would be a breach of our ethics approval and participant trust. 

Response: The supporting information has been added at the end of the manuscript, after the reference list (lines 946-947). On page 7, line 118, of our manuscript, we have updated that information.

Response: We have checked all of the references, and no papers have been retracted. We have removed all of the URLs and we have included links to web pages or resources from national government websites. 

Reviewers' comments:

Reviewer's Responses to Questions

Comments to the Author

1. Is the manuscript technically sound, and do the data support the conclusions?

Reviewer #1: Yes

2. Has the statistical analysis been performed appropriately and rigorously?

Reviewer #1: N/A

3. Have the authors made all data underlying the findings in their manuscript fully available?

The PLOS Data policy requires authors to make all data underlying the findings described in their manuscript fully available without restriction, with rare exception (please refer to the Data Availability Statement in the manuscript PDF file). The data should be provided as part of the manuscript or its supporting information or deposited to a public repository. For example, in addition to summary statistics, the data points behind means, medians and variance measures should be available. If there are restrictions on publicly sharing data—e.g. participant privacy or use of data from a third party—those must be specified.

Reviewer #1: Yes

4. Is the manuscript presented in an intelligible fashion and written in standard English?

Reviewer #1: Yes

5. Review Comments to the Author

Reviewer #1: I think it is very well written. Please address the following minor points.

1. Lines 238-240 suggest that having experienced staff reduces the risk of falls. Specifically, what is the difference between younger and more experienced staff? I would like to see a few sentences in the discussion to include perspectives from previous studies.

Response: Thank you for bringing this to our attention. We have discussed this in detail on pages 31 and 32, lines 649-662.

2. line573

Topic 5 is probably a mistake for topic 6.

Response: Thank you for pointing this out. This has been corrected to “Theme 6” on page 28, line 573.

---

## [Decision Letter · Decision Letter 1]

7 Jun 2024

PONE-D-23-40350R1"Brief but often": Staff perspectives on implementing falls prevention activities in long-term care facilities in older persons: a qualitative studyPLOS ONE

Dear Dr. ALBASHA,

Thank you for submitting your manuscript to PLOS ONE. After careful consideration, we feel that it has merit but does not fully meet PLOS ONE’s publication criteria as it currently stands. Therefore, we invite you to submit a revised version of the manuscript that addresses the points raised during the review process.

We look forward to receiving your revised manuscript.

Kind regards,

Nabeel Al-Yateem, PhD

Academic Editor

PLOS ONE

Journal Requirements:

Reviewers' comments:

Reviewer's Responses to Questions

**Comments to the Author**

1. If the authors have adequately addressed your comments raised in a previous round of review and you feel that this manuscript is now acceptable for publication, you may indicate that here to bypass the “Comments to the Author” section, enter your conflict of interest statement in the “Confidential to Editor” section, and submit your "Accept" recommendation.

Reviewer #1: All comments have been addressed

Reviewer #2: (No Response)

2. Is the manuscript technically sound, and do the data support the conclusions?

Reviewer #1: Yes

Reviewer #2: Yes

3. Has the statistical analysis been performed appropriately and rigorously? 

Reviewer #1: Yes

Reviewer #2: Yes

4. Have the authors made all data underlying the findings in their manuscript fully available?

Reviewer #1: Yes

Reviewer #2: Yes

5. Is the manuscript presented in an intelligible fashion and written in standard English?

Reviewer #1: Yes

Reviewer #2: Yes

6. Review Comments to the Author

Reviewer #1: The authors have responded appropriately and effectively to the comments.

No further revision required.

Reviewer #2: Clearly written and scholarly article. Only a few points need to be addressed:

1. The overall length of this manuscript is far too long - please reduce by at least 300 words.

2. The title is excessively long and complication - I had to read it several times. Please do a major revision of the title to make it clear and concise rather than long and descriptive.

3. The methods section is excessively long and I am concerned that some sections on qualitative methods seem to be very similar to the original source - please run the manuscript thorough iThenticate (or similar) and reduce text so you are not repeating other authors words so much. Just because you give a reference does not mean that you can have large chucks of very similar text.

4. The Discussion could cite and discuss the learnings from the hospital falls literature a bit more as these are highly relevant and would bring your review up to date. Some refs to consider include:

Morris, M.. (2024). Preventing hospital falls: feasibility of care workforce redesign to optimise patient falls education. Age and Ageing, 53(1), 9 pages. doi:10.1093/ageing/afad250

Montero-Odasso, M., Van Der Velde, N., Martin, F. C., Petrovic, M., Tan, M. P., Ryg, J., . . . Gac Espinola, H. (2022). World guidelines for falls prevention and management for older adults: a global initiative. Age and Ageing, 51(9). doi:10.1093/ageing/afac205

Morris, M. E.,(2022). Interventions to reduce falls in hospitals: a systematic review and meta-analysis.. Age and Ageing, 51(5), 12 pages. doi:10.1093/ageing/afac077

Heng, H.,. (2020). Hospital falls prevention with patient education: a scoping review. BMC Geriatrics, 20, 12

5. Please update the ref list to include more recent refs.

7. PLOS authors have the option to publish the peer review history of their article (what does this mean?). If published, this will include your full peer review and any attached files.

Reviewer #1: No

Reviewer #2: No

---

## [Author Response · Author response to Decision Letter 1]

12 Jul 2024

Journal Requirements:

 Response: We have checked all of the references, and no papers have been retracted. We have removed all of the URLs, and we have included links to web pages or resources from national government websites. We have reviewed and removed all the old papers from the reference list except those that we considered important to our paper. 

Reviewers' comments:

Reviewer's Responses to Questions

Comments to the Author

1. If the authors have adequately addressed your comments raised in a previous round of review and you feel that this manuscript is now acceptable for publication, you may indicate that here to bypass the “Comments to the Author” section, enter your conflict of interest statement in the “Confidential to Editor” section, and submit your "Accept" recommendation.

Reviewer #1: All comments have been addressed

Reviewer #2: (No Response)

2. Is the manuscript technically sound, and do the data support the conclusions?

Reviewer #1: Yes

Reviewer #2: Yes

3. Has the statistical analysis been performed appropriately and rigorously?

Reviewer #1: Yes

Reviewer #2: Yes

4. Have the authors made all data underlying the findings in their manuscript fully available?

Reviewer #1: Yes

Reviewer #2: Yes

5. Is the manuscript presented in an intelligible fashion and written in standard English?

Reviewer #1: Yes

Reviewer #2: Yes

6. Review Comments to the Author

Reviewer #1: The authors have responded appropriately and effectively to the comments.

No further revision required.

Reviewer #2: Clearly written and scholarly article. Only a few points need to be addressed:

1. The overall length of this manuscript is far too long - please reduce by at least 300 words.

Response: We have reduced the total word count of the text by around 650 words, as it originally contained 8,109 words, and now it has (7,461 words) 

2. The title is excessively long and complication - I had to read it several times. Please do a major revision of the title to make it clear and concise rather than long and descriptive.

Response: We have revised the title to make it clearer 

3. The methods section is excessively long and I am concerned that some sections on qualitative methods seem to be very similar to the original source - please run the manuscript thorough iThenticate (or similar) and reduce text so you are not repeating other authors words so much. Just because you give a reference does not mean that you can have large chucks of very similar text.

Response: We checked for any plagiarism and could not find any. We are confident that our manuscript is well-written and is free of plagiarism. Our methods section was written in depth, as we followed the COREQ guideline, and most of the items focused on the methods. In accordance with the guideline, we have reported all of our procedures in detail. However, we have acknowledged that our methods section was too long, as it contained 1,205 words, so we reduced it to (908 words). 

4. The Discussion could cite and discuss the learnings from the hospital falls literature a bit more as these are highly relevant and would bring your review up to date. Some refs to consider include:

Morris, M.. (2024). Preventing hospital falls: feasibility of care workforce redesign to optimise patient falls education. Age and Ageing, 53(1), 9 pages. doi:10.1093/ageing/afad250

Montero-Odasso, M., Van Der Velde, N., Martin, F. C., Petrovic, M., Tan, M. P., Ryg, J., . . . Gac Espinola, H. (2022). World guidelines for falls prevention and management for older adults: a global initiative. Age and Ageing, 51(9). doi:10.1093/ageing/afac205

Morris, M. E.,(2022). Interventions to reduce falls in hospitals: a systematic review and meta-analysis.. Age and Ageing, 51(5), 12 pages. doi:10.1093/ageing/afac077

Heng, H.,. (2020). Hospital falls prevention with patient education: a scoping review. BMC Geriatrics, 20, 12

Response: Thank you for the recommendations you provided. The resources are beneficial since they offer significant knowledge on the prevention of falls among older people. However, our primary objective is to prevent falls among older people residing in long-term care facilities. Consequently, we will not include all these references, as they pertain to the context of acute care (hospitals) and are not relevant to our study. We have included the SR by Morris et al as a usual reference for readers. Also, we already had referenced the World Guidelines for Falls Prevention (the second reference from Montero-Odasso et al., as you suggested) (reference number 7 (previously 47) in our manuscripts).

5. Please update the ref list to include more recent refs.

Response: Our reference list has been reviewed and updated with a few new papers, and older papers have been removed except those that are important for comparing our results for context.

---

## [Decision Letter · Decision Letter 2]

26 Aug 2024

Staff perspectives on fall prevention activities in long-term care facilities for older residents: "brief but often" staff education is key

PONE-D-23-40350R2

Dear Dr. Albasha,

We’re pleased to inform you that your manuscript has been judged scientifically suitable for publication and will be formally accepted for publication once it meets all outstanding technical requirements.

Kind regards,

Mohammad Jamil Rababa

Academic Editor

PLOS ONE

Additional Editor Comments (optional):

Reviewers' comments:

Reviewer's Responses to Questions

**Comments to the Author**

1. If the authors have adequately addressed your comments raised in a previous round of review and you feel that this manuscript is now acceptable for publication, you may indicate that here to bypass the “Comments to the Author” section, enter your conflict of interest statement in the “Confidential to Editor” section, and submit your "Accept" recommendation.

Reviewer #1: All comments have been addressed

Reviewer #2: All comments have been addressed

2. Is the manuscript technically sound, and do the data support the conclusions?

Reviewer #1: Yes

Reviewer #2: Yes

3. Has the statistical analysis been performed appropriately and rigorously? 

Reviewer #1: Yes

Reviewer #2: Yes

4. Have the authors made all data underlying the findings in their manuscript fully available?

Reviewer #1: Yes

Reviewer #2: Yes

5. Is the manuscript presented in an intelligible fashion and written in standard English?

Reviewer #1: Yes

Reviewer #2: Yes

6. Review Comments to the Author

Reviewer #1: I think it is well written. I think it is more compact and easier to read than previous posts. Thank you.

Reviewer #2: The revised manuscript is much improved, thank you. My only question is should you call them "residents" rather than "patients" as it is aged care.

7. PLOS authors have the option to publish the peer review history of their article (what does this mean?). If published, this will include your full peer review and any attached files.

Reviewer #1: No

Reviewer #2: No

---

## [Editor Report · Acceptance letter]

29 Aug 2024

PONE-D-23-40350R2 

PLOS ONE

Dear Dr. ALBASHA, 

I'm pleased to inform you that your manuscript has been deemed suitable for publication in PLOS ONE. Congratulations! Your manuscript is now being handed over to our production team.

Kind regards, 

on behalf of

Dr. Mohammad Jamil Rababa 

Academic Editor

PLOS ONE